# Effect of Fermentation Parameters on Natto and Its Thrombolytic Property

**DOI:** 10.3390/foods10112547

**Published:** 2021-10-22

**Authors:** Yun Yang, Guangqun Lan, Xueyi Tian, Laping He, Cuiqin Li, Xuefeng Zeng, Xiao Wang

**Affiliations:** 1Key Laboratory of Agricultural and Animal Products Store & Processing of Guizhou Province, Guizhou University, Guiyang 550025, China; 18984280534@163.com (Y.Y.); langq@ssjf49.com (G.L.); 14785719587@163.com (X.T.); licuiqin2345@163.com (C.L.); heiniuzxf@163.com (X.Z.); wangzi8903@126.com (X.W.); 2College of Liquor and Food Engineering, Guizhou University, Guiyang 550025, China; 3School of Chemistry and Chemical Engineering, Guizhou University, Guiyang 550025, China

**Keywords:** natto, nattokinase, combination fermentation, thrombolytic property

## Abstract

Natto is a popular food because it contains a variety of active compounds, including nattokinase. Previously, we discovered that fermenting natto with the combination of *Bacillus subtilis* GUTU09 and *Bifidobacterium animalis subsp. lactis* BZ25 resulted in a dramatically better sensory and functional quality of natto. The current study further explored the effects of different fermentation parameters on the quality of natto fermented with *Bacillus subtilis* GUTU09 and *Bifidobacterium* BZ25, using Plackett–Burman design and response surface methodology. Fermentation temperature, time, and inoculation amount significantly affected the sensory and functional qualities of natto fermented with mixed bacteria. The optimal conditions were obtained as follows: soybean 50 g/250 mL, triangle container, 1% sucrose, *Bacillus subtilis* GUTU09 to *Bifidobacterium* BZ25 ratio of 1:1, inoculation 7%, fermentation temperature 35.5 °C, and fermentation time 24 h. Under these conditions, nattokinase activity, free amino nitrogen content, and sensory score were increased compared to those before optimization. They were 144.83 ± 2.66 FU/g, 7.02 ± 0.69 mg/Kg and 82.43 ± 5.40, respectively. The plate thrombolytic area and nattokinase activity both increased significantly as fermentation time was increased, indicating that the natto exhibited strong thrombolytic action. Hence, mixed-bacteria fermentation improves the taste, flavor, nattokinase activity, and thrombolysis of natto. This research set the groundwork for the ultimate manufacturing of natto with high nattokinase activity and free amino nitrogen content, as well as good sensory and thrombolytic properties.

## 1. Introduction

People’s living standards have gradually risen in recent years, and the incidence of thrombosis in blood arteries has increased year after year [1]. Acute myocardial infarction, hypertension, and stroke, for example, are commonly linked to excessive fibrin deposition in blood arteries [2]. Meanwhile, the prevalence of cardiovascular and cerebrovascular diseases has skyrocketed. As a result, the importance of healthy products and foods in the prevention of cardio-cerebrovascular diseases has grown. In recent years, a natural fermentation enzyme known as nattokinase has been a prominent study issue among those nutritious foods. Natto is a soybean that has a distinct flavor and texture, which is made by fermenting soybeans with *Bacillus subtilis*. It originated in ancient China and evolved into a new form of fermented food after being exported to Japan during the Tang Dynasty. The fermentation of natto produces a number of bioactive components, including nattokinase, daidzein, phytosterols, superoxide dismutase, and several biologically active peptides, in addition to preserving the nutritional value of soybeans [3]. Nattokinase is a fibrinolytic enzyme released by the *Bacillus subtilis* natto bacteria during the fermentation process [4]. Nattokinase has better thrombolytic activity and selectivity when compared to other fibrinolytic enzymes [5]. Furthermore, after oral administration, nattokinase is rapidly absorbed across the gastrointestinal tract [6], causing fibrinolysis. Thus, it is evaluated as a possible clot-dissolving agent for the treatment and prevention of cardiovascular disease [7].

Natto has limited public awareness due to its distinct flavor produced by *Bacillus subtilis*. The dispute continues to focus on natto’s pungent ammonia smell and flavor, as well as a lack of convincing proof that it reduces the risk of thrombotic disorders. Overall, the higher the nattokinase activity, free amino nitrogen content and sensory score, the better the natto’s health advantages and acceptance. As a result, increasing the content of multiple biological activities, and improving the sensory score and flavor of natto for the general population are demanded. *Bifidobacterium* may produce lactic acid and acetic acid, which can cover the ammonia and its distinct taste, as well as other functional compounds; thus, adding additional *Bifidobacterium* to the natto may improve the flavor and quality.

We previously isolated nattokinase-producing *Bacillus subtilis* GUTU09 (CCTCC M 2021641) which may be used to produce natto and *Bifidobacterium* BZ25 (CGMCC NO.10225) [8]. However, the two bacteria strains’ fermentation parameters for producing natto were not investigated. As a result, the goal of the current study was to apply Plackett–Burman design (PB) in conjunction with Box–Behnken design (BBD) to investigate the effect of fermentation parameters to produce high-quality natto. In addition, the natto’s thrombolytic properties were examined.

## 2. Materials and Methods

### 2.1. Materials

The *Bacillus subtilis* GUTU09 (B9) and *Bifidobacterium animalis subsp. lactis* BZ25 were strains with excellent performance screened by our laboratory from Guizhou local characteristic food [9,10]. Thrombin (1000U) and bovine fibrinogen were purchased from Solarbio Science and Technology Co. (Beijing, China). Other culture media and analytical chemicals were purchased from Sinopharm Chemical Reagent Co., Ltd. (Shanghai, China).

### 2.2. Seed Preparation

BZ25 was inoculated in MRS medium (natural pH) and anaerobically cultured at 37 °C for 48 h [11]. B9 was cultivated in the liquid seed medium containing 10 g/L glucose, 5 g/L yeast extract, 10 g/L beef extract, and 5 g/L NaCl with pH of 7.0–7.5 at 37 °C and 180 r/min for 18 h. After that, the cells were extracted by centrifugation at 4000× *g* (TGL20M High-speed freezing centrifuge, Maijiassen instrument, and equipment Co., Ltd., Changsha, China) for 10 min and resuspended in a small amount of sterile physiological saline. The concentrations of BZ25 and B9 were then determined by anaerobic or aerobic culture in either MRS agar plates (MRS agar medium was prepared by adding 20 g of agar per liter MRS broth medium) or B9 liquid medium (B9 agar medium was prepared by adding 20 g of agar per liter B9 liquid medium) at 37 °C for 48 h. Subsequently, its cell concentrations were diluted to 1 × 10^8^ CFU/mL with physiological saline [12].

### 2.3. Natto Preparation

Soybeans from Heilongjiang, China, were soaked overnight in 20 °C water. Wet beans (50 g) were sterilized at 121 °C for 20 min, then cooled to 25 °C in a 250 mL conical flask. Soybeans were inoculated by 4% of BZ25(1 × 10^8^ CFU/mL) and 4% of B9 (1 × 10^8^ CFU/mL) and fermented under static conditions (250 mL flask is sealed with eight layers of gauze) for 24 h at 37 °C. The fermented soybeans were then ripened for 24 h at 4 °C before being used to make natto.

### 2.4. One-Factor-at-a-Time Experiments

Using pH (initial pH was 7.5), nattokinase activity, free amino nitrogen, and sensory scores as indicators, the effects of NaCl, sucrose, the ratio of B9 to BZ25 strains, inoculation amount, fermentation temperature, fermentation time, and after-ripening time on natto were investigated by one-factor-at-a-time experiments.

### 2.5. PB Design

NaCl (A), sucrose (B), fermentation temperature (D), fermentation time (E), strain ratio (G), inoculation amount (J), and after-ripening time (L) were chosen for PB design based on the findings of the single-factor experiment. Seven independent and four dummy variables (C, F, H, and K) were examined at two levels in total (Table 1).

### 2.6. BBD

The results of the PB design demonstrate that the most critical parameters influencing soybean fermentation were fermentation time, fermentation temperature, and inoculation amount. We applied BBD and used nattokinase activity, free amino nitrogen, and sensory scores as response values. Each of the parameters is listed in Table 2.

### 2.7. pH Measurement and Extraction of Crude Enzyme Solution

Natto (10 g) was dissolved in 90 mL deionized water, homogenized for 30 s, and extracted for 24 h at 4 °C. A pH meter was used to determine the pH of the suspension. The supernatant was kept at 4 °C after centrifugation at 16,000× *g* (TGL20M High-speed freezing centrifuge, Maijiassen instrument, and equipment Co., Ltd., Changsha, China) for 10 min for nattokinase activity determination.

### 2.8. Free Amino Nitrogen Contents (FANs)

The natto suspension was centrifuged at 1800× *g* for 10 min after being extracted at 4 °C for 24 h, and the supernatant was kept at 4 °C for later use. The method described by Lu et al. [13] was used to determine the free amino nitrogen.

### 2.9. Nattokinase (NK) Activity Assay

The fibrinolytic activity of nattokinase was measured using the V. Deepak et al. [14] technique.

### 2.10. Sensory Properties

The sensory properties of natto were evaluated by the method of Feng et al. [15] with some modifications. A sensory evaluation team of 15 teachers and graduate students with sensory evaluation experience in food-related courses was assembled. The sensory evaluation of natto was carried out in a well-ventilated food lab with plenty of light and room. Appearance, viscosity, flavor, mouthfeel, and chewiness are the key determinants of sensory properties. The appearance of the product was evaluated by color intensity, color consistency, brightness, and gloss. The length, density, and adhesion of natto to the chopsticks were used to determine stickiness. The flavor of natto was evaluated by fragrance, ammonia, and beany flavor. It was expected that the flavor would be mild or somewhat sour. The overpowering bitterness was unappealing. Teeth’s feedback to the softness, hardness, stickiness, and smoothness of natto comprised chewiness. Rating values of 1 to 5 were utilized for independent evaluation of these five sensory properties (5 = like very much, 4 = like more, 3 = average, 2 = dislike very much, 1 = dislike very much). Finally, each sensory rating was multiplied by a factor of four to provide an index score. The greater the index score, the higher the natto’s quality [16].

### 2.11. Thrombolytic Effect

The area and diameter of the dissolution circle in the fibrin plate were used to evaluate the thrombolytic action of crude enzyme solution using the agarose fibrinogen plate method as described by Gao [10].

### 2.12. Anticoagulant Activity

The anticoagulant activity was determined by measuring the inhibition of fibrinogen to fibrin conversion, using Wei’s [17] approach with minor changes. The details are as follows: First, 0.3 mL of fibrinogen solution (7.2 mg/mL) was combined with 1.2 mL of Tris–HCl (50 mM, pH 7.8) buffer in a colorimetric cuvette with 1.0 mL of the diluted sample solution. After that, 0.1 mL of thrombin solution (20 U/mL) was added, the mixture was incubated at 37 °C for a while, and the absorbance was measured at 405 nm (UV-Vis Spectrophotometer TU-1810PC Pullout General Instrument Co., Ltd., Beijing, China.). As a control, no sample solution was used. The anticoagulant activity of natto is estimated by formula (1) below:(1)$$\mathrm{Activity}\;\mathrm{of}\;\mathrm{coagulation}\;\mathrm{inhibitory}\;\%=\left(1-OD_{\mathrm{sample}}\right)/OD_{\mathrm{Control}}\times 100\%$$

### 2.13. Statistical Analysis

All of the tests were performed in triplicate. The data are presented as mean ± standard deviation (SD). One-way ANOVA was performed using SPSS 19.0 (SPSS Inc., Chicago, IL, USA), and differences were determined using Tukey’s HSD test, with *p* values < 0.05 considered statistically significant. Design-Expert 8.0.6 software (Stat-Ease, Inc., Minneapolis, MN, USA) was used for the PB test and BBD and for their data analysis, and Origin 2018 software (OriginLab, Northampton, MA, USA) was used for drawing the graph.

## 3. Results

### 3.1. Effect of NaCl on Natto

Figure 1(a_1_,b_1_,c_1_) show the effect of NaCl on natto. The sensory qualities of natto were significantly affected by different NaCl concentrations. The highest sensory score was earned by 1% NaCl. The sensory scores, on the other hand, showed no significant difference (*p* > 0.05) between 0% and 1%. However, as the salt concentration increased above 1%, the sensory score rapidly decreased, and the difference was extremely significant (*p* < 0.01). Figure 1(b_1_,c_1_) demonstrate that as NaCl levels rise, pH, nattokinase activity, and free amino nitrogen levels fall, with the highest indices appearing in the control group (NaCl equal to 0).

### 3.2. Effect of Sucrose on Natto

Figure 1(a_2_,b_2_,c_2_) reveals that the amount of sucrose has a highly significant (*p* < 0.01 obtained by one-way ANOVA analysis) impact on the sensory score of natto. The sensory score was highest when the sugar concentration was 1–2%. When the sugar level in the flask fermentation environment reached over 2%, the higher sugar content made the mixed-bacteria fermentation produce more acid. This is mainly due to the fact that O_2_ would become almost depleted in the solid fermentation of soybean, and this favored the growth of the *Bifidobacterium* and inhibited the strictly aerobic *Bacillus* strain. The extra acid made the natto taste and flavor worse, as well as making the bean difficult to chew and increasing the stringiness viscosity. When the sugar content approached 3%, the sensory score, nattokinase activity, and free amino nitrogen levels all decreased significantly (*p* < 0.05).

### 3.3. Effect of Strain Proportion on Natto

Figure 2(a_1_,b_1_,c_1_) depict the influence of strain ratio on natto fermentation. The ratio of strains did not affect free amino nitrogen or sensorial properties (*p* > 0.05). The pH, on the other hand, rose initially and subsequently fell as the strain ratio rose. When the strain ratios were 2:1, 1:1, and 1:2, there was no significant change in pH (*p* > 0.05). Furthermore, when the ratio increased, nattokinase activity reduced progressively but not significantly.

### 3.4. Effect of Inoculation Amount on Natto

Figure 2(a_2_,b_2_,c_2_) show the effect of inoculation amount on fermented natto. The activity of nattokinase and free amino nitrogen concentration increased and then reduced when the inoculation amount was increased, which was consistent with earlier studies [18]. The highest levels of nattokinase activity, sensory score, and free amino nitrogen were attained when the inoculation amount reached 6–8%. When the inoculation concentration was less than 6% or higher than 8%, nattokinase activity, free amino nitrogen, and sensory score decreased.

### 3.5. Effect of Fermentation Temperature on Natto

As demonstrated in Figure 3(a_1_,b_1_,c_1_), the nattokinase activity was highest at 40 °C; however, the sensory score was not the best at this temperature. In terms of flavor, texture, stringiness, and appearance, the temperature has a bigger impact on these sensory characteristics. When the temperature was between 31–34 °C, the bacteria grew slower and had an impact on various biochemical reactions throughout the fermentation. When the temperature was higher than 37 °C, the sensory score gradually decreased, and the activity of nattokinase first increased and eventually decreased. In addition, as the temperature rose, the pH increased as well (Figure 3(b_1_)), and the amount of free amino nitrogen also increased.

### 3.6. Effect of Fermentation Time on Natto

Figure 3(a_2_,b_2_,c_2_) show the effect of fermentation time on fermented natto. Natto hade a high sensory score when fermented for 18 or 24 h. The color of the natto darkened as the fermentation time progressed, and the ammonia smell became stronger. Excessive fermentation softened the bean grains, lowering the sensory score. However, the activity of nattokinase and the quantity of free amino nitrogen increased as the fermentation duration increased. 

### 3.7. Effect of After-Ripening Time on Natto

Figure 4a–c depicts the influence of after-ripening time on natto. Natto scored a higher sensory score of 2–3 days after ripening time. However, extending the after-ripening time reduces the product quality and freshness, lowering the sensory score. The pH changes slightly during the after-ripening process, suggesting the dynamics of the composition of natto. The free amino nitrogen steadily rose as the after-ripening period increased, demonstrating that the protease was still hydrolyzing soybean protein even at low temperatures. Despite the fact that the overall trend in nattokinase was increasing, it seemed that there was a downward trend in the end, which is consistent with prior observations [19].

### 3.8. Plackett–Burman (PB) Design

According to the analysis of the single factor test, the Design-Expert software was used to carry out the PB design (Table 1, Appendix A), and the key factors that significantly affected the fermentation of natto were screened out.

The analysis of variance response values with nattokinase activity, free amino nitrogen content, and sensory scores are shown in Table 3. The *p*-values of the three ANOVA models are all less than 0.05, indicating that the three ANOVA models are significant and have a good degree of fit. We discovered that NaCl, sucrose, fermentation temperature, and fermentation duration all had significant (*p* < 0.05) influences on nattokinase activity. The following is the sequence in which factors impact nattokinase: NaCl > sucrose > fermentation temperature > fermentation time. The amount of NaCl and sucrose, fermentation temperature, and fermentation time had an extremely significant (*p* < 0.01) effect on the content of free amino nitrogen. The following is the sequence in which factors impact the production of free amino nitrogen: NaCl > sucrose > fermentation time > fermentation temperature. Fermentation temperature, fermentation period, and inoculum amount all had an impact on the sensory score (*p* < 0.05). The following is the sequence in which factors affected sensory score: inoculum amount > fermentation time > fermentation temperature.

In order to obtain products with good sensory characteristics, as a result, three factors were chosen for the next response surface analysis: fermentation time (E), fermentation temperature (D), and inoculation amount (J).

### 3.9. Response Surface Analysis by Box–Behnken Design (BBD)

The result of the BBD is shown in Appendix A. All data are fitted to a second-order polynomial model (Y_1_, Y_2_, Y_3_). The significance of these models is expressed as a low *p* value (Appendix A, nattokinase activity *p* = 0.0174; free amino nitrogen content *p* = 0.0041; sensory score *p* = 0.0004), and the lack of fit of these models was not significant, suggesting the models as being credible. When the R^2^ value is greater than 0.8, the response surface model is generally considered appropriate [20]. The higher R^2^ (R^2^ > 0.8) of the three responses indicates a high correlation between the predicted value and the actual value.
(2)$$Y_{1}=132.58+5.08\times A-1.16\times B-3.04\times C+1.79\times \mathrm{AB}-1.29\times \mathrm{AC}+0.78\times \mathrm{BC}+7.27\times A^{2}-3.28\times B^{2}+2.7\times C^{2}$$
(3)$$Y_{2}=5.51+1.71\times A+0.65\times B-0.28\times C+0.89\times \mathrm{AB}+0.12\times \mathrm{AC}-0.24\times \mathrm{BC}+0.94\times A^{2}-0.73\times B^{2}+0.59\times C^{2}$$
(4)$$Y_{3}=85.28-3.15\times A-2.18\times B+1.83\times C-3.90\times \mathrm{AB}-1.30\times \mathrm{AC}+2.75\times \mathrm{BC}-8.76\times A^{2}+0.085\times B^{2}-6.22\times C^{2}$$

According to the model equation Y_1_, the linear term A, cross-product terms AB and BC, and quadratic terms A^2^–C^2^ had a positive effect on the nattokinase activity, while the linear terms B and C, cross-product term AC, and quadratic term B^2^ showed a negative effect. Appendix A shows that only the linear terms of fermentation time (A), inoculation volume (C), and quadratic terms of A^2^ (*p* < 0.05) were substantially linked with NK production.

The 3D surfaces of the interactive effects of fermentation time (A), fermentation temperature (B), and inoculation volume (C) on the NK are illustrated in Figure 5. Figure 5(A_1_) shows with the increase in temperature and time, the activity of nattokinase decreased at first and then increased. As shown in Figure 5(B_1_), with the extension of the fermentation time, the activity of nattokinase showed an increasing trend. This is because the longer the fermentation time, the more time the bacteria will have to grow and ferment, resulting in more nattokinase being produced by the bacteria. The combined effect of inoculum amount and fermentation temperature revealed that nattokinase activity was affected by fermentation temperatures between 31 and 37 °C, but the effect was minor (Figure 5(C_1_)).

Similar to nattokinase activity, Appendix A shows that the linear terms A-B, cross-product terms AB and AC, and quadratic terms A^2^-C^2^ had a significantly (*p* < 0.05) positive influence on the free amino nitrogen (FAN)content, while the linear terms C, cross-product terms BC and quadratic terms B^2^ presented a significantly (*p* < 0.05) negative effect (equation Y_2_).

The level of FAN increased as the temperature and fermentation time increased (Figure 5(A_2_)). The fermentation period influences the accumulation of products. The free amino nitrogen concentration first declined and subsequently increased as the amount of inoculation and fermentation time increased (Figure 5(B_2_)).

Appendix A shows that the linear term C, cross-product term BC, and quadratic term B^2^ had a significantly (*p* < 0.05) positive influence on the sensory score, while the linear terms A-B, cross-product terms AB and AC, and quadratic terms A^2^-C^2^ presented a significantly (*p* < 0.05) negative effect (equation Y_3_). The sensory score of natto climbed as fermentation time, fermentation temperature, and inoculum amount increased, reaching a maximum. Then, it began to decline as these three elements continued to rise (Figure 6).

### 3.10. Determination and Verification of Natto Process

After response surface optimization, according to the second-order model formula, the theoretical optimal conditions were: sucrose addition was 1%, inoculation amount was 6.96%, fermentation temperature was 35.44 °C, and fermentation time was 24 h. Under these conditions, the predicted value of nattokinase activity was 132.56 FU/g, the sensory score predictive value was 81.21, and the free amino nitrogen content was 4.34 mg/kg. The preceding parameters were slightly modified as follows: sucrose at 1%, inoculation amount at 7%, fermentation temperature at 35.5 °C, and fermentation period at 24 h, taking into account the feasibility of the actual operation.

In order to verify the effectiveness and feasibility of the model, three replicate tests were performed under the optimal culture conditions. The results show that the activity of nattokinase was 144.83 ± 2.66 FU/g, the sensory score was 82.43 ± 5.40, and the free amino nitrogen content was 7.02 ± 0.69 mg/Kg. All of the above values were in line with the expected outcomes. As a result, we came to the conclusion that the model could accurately predict the real fermentation data for natto.

### 3.11. Thrombolytic Activity, Fibrin Degradation Ability, and Anticoagulant Activity

The thrombolytic activities of natto were investigated utilizing the agarose–fibrinogen plate method under the abovementioned ideal circumstances. The fibrinolysis action of natto extract is shown in Figure 7A. The diameter of the dissolution circle given in Table 4 was measured under aseptic conditions. The area of the dissolution circle was calculated according to the diameter (π = 3.14). The area of thrombolysis is the area of the dissolution circle minus the area of the central pore (1.98 ± 0.39 mm ^2^).

As shown in Table 4, as the fermentation time rose, the corresponding diameter of the dissolving circle increased in proportion to the thrombolytic area, and the thrombolysis became more obvious.

Figure 7B depicts the variations in nattokinase activity throughout single-strain and mixed-strain fermentation. The activity of nattokinase gradually increased as the fermentation period was extended. The maximum activity of nattokinase was found in a 36 h fermentation of mixed bacteria, with an enzyme activity of 161 ± 1.96 FU/g recorded. When comparing the two fermentation methods, single-bacteria fermentation had higher natto enzyme activity before 18 h, and equal the enzyme activity at 18 h. After 18 h, the mixed-bacteria fermentation had higher enzyme activity than single-bacteria, which was similar to the fibrinolytic activity indicated before. Finally, the mixed-bacteria fermentation method produced higher nattokinase activity, more free amino nitrogen, and a superior sensory evaluation. As a result, it is a nutritious food that is readily available to the general public.

The antithrombotic process was recorded when the crude enzyme extract of natto (151.06 ± 2.34 FU/g) was diluted 50 times, 90 times, and 130 times with no added sample as a control (Figure 7C). The control group reached the maximum fibrin concentration (OD_405_ = 1.107) at 20 min. When the dilution ratio was 130 times, the formation of fibrin was inhibited and the OD_405_ was 0.494 at 46 min, and fibrin was completely degraded within 100 min. In contrast, the formation of fibrin in a 90-fold diluted sample solution was inhibited at 38 min, and fibrin was almost completely degraded within 60 min. The inhibitory effect, on the other hand, was better and the time was shorter. When the dilution ratio was 50 times, there was no significant change in the curve, indicating that it had strong anticoagulant activity.

## 4. Discussion

The formation of a blood clot in a blood vessel is one of the main causes of cardiovascular diseases [5]. Blood clots are formed from fibrinogen via proteolysis by thrombin, and can be hydrolyzed by plasmin to avoid thrombosis in blood vessels [21]. Natural products with anticoagulant activity can hinder the conversion of fibrinogen into fibrin and play a role in the final stage of the anticoagulation process [22]. Natto offers several health benefits, with nattokinase being the most essential functional component. Nattokinase, a serine protease secreted by *Nattobacillus*, has high fibrinolytic activity and has been proved to be an effective thrombolytic in vitro and in vivo [23]. The health benefits of natto have always attracted people’s interest [24], and the improvement in its quality has always been a goal that people pursue, including improving the flavor and taste of natto, and increasing the content of functional ingredients. The combination of *Bacillus subtilis* GUTU09 and *Bifidobacterium* BZ25 (mixed bacteria) was utilized to ferment natto in this study. Sensory score, pH, nattokinase, and free amino nitrogen concentration were used as indexes to investigate the impact of various variables on the sensory and functional quality of fermented natto.

The indexes of natto decreased with the addition of NaCl (Figure 1(b_1_,c_1_)). It may be due to the inhibition of some enzyme activities of the two strains under high- and medium-salt conditions. Yue [25] et al. found that histidine decarboxylase was completely inactivated in high- and medium-salt conditions, but remained active when the salt concentration decreased. Meanwhile, because of health concerns associated with high salt intakes, the demand for low-salt foods has surged. High salt intake is linked to diseases such as hypertension and renal impairment [26]. As a result, excessive salt should not be used in the fermentation process. Hypertension can be prevented and controlled by eating low-salt foods [27]. It is very beneficial to people with coronary heart disease and hypertension, and it meets the nutritional needs of modern society.

The effect of raising the sucrose content differs from that of raising the salt concentration (Figure 1). Adding about 1–2% sugar will improve the sensory characteristics. The stringiness, flavor, taste, and chewiness of natto are all affected by the sugar concentration. Despite the presence of oligosaccharides and polysaccharides in the steamed soybean, it lacks a carbon source that BZ25 can utilize directly. Increasing the carbon source sugar that the microbes can consume directly results in accelerated growth and metabolism. Our findings are in line with a previous study by Wu et al. [28]. On the one hand, more sucrose provides a better environment for BZ25 to grow, and acid production as a by-product of the microbe increases rapidly, lowering the pH throughout the fermentation process (Figure 1(b_2_)) and improving the texture and aroma of natto. Adding sucrose for B9, on the other hand, causes rapid fermentation and the production of numerous enzymes to hydrolyze the macromolecular components in soybeans. This not only promotes BZ25 proliferation but also accumulates a significant number of fermentation products.

In the same fermentation environment, a high strain ratio not only prevents the synergy between the strains from working, but also lowers the quality of the fermented product due to nutrient competition between the two strains [29]. A high strain ratio of BZ25 to GUTU09 also produced much acid (Figure 2(b_1_)), which led to the decline in sensory quality. The reason may be that the internal oxygen content in the solid fermentation was generally low, and increasing the ratio of *Bifidobacterium* BZ25 to *Bacillus subtilis* GUTU09 can enhance the resistance of *Bifidobacterium* to a small amount of oxygen and make it easy to propagate rapidly and produce more acid in the fermentation. Therefore, it is not beneficial to natto fermentation if the ratio of strains GUTU09 to BZ25 is too large or too small. As a result, the suitable strain ratio was approximately 2:1 or 1:1 (Figure 2(a_1_,b_1_,c_1_)).

The number of initial bacteria in the fermentation process is determined by the amount of inoculation, and the appropriate starting bacteria concentration can shorten the strain’s growth and proliferation period, allowing it to enter the fermentation stage sooner. An inoculation quantity ranging from 6 to 8% (Figure 2(a_2_,b_2_,c_2_)) is more appropriate; too high or too low inoculation may impair the natto quality. If the initial bacteria concentration is lower, the time needed for reproduction will be longer and the fermentation speed will be slower, which is not conducive to the accumulation of fermentation products. Higher inoculation concentrations, on the other hand, resulted in a strong and speedy bacterial metabolism, requiring the cell to consume the majority of the nutrients in order to continue its rapid growth and hinder the accumulation of fermentation products. When a large amount of metabolic waste is produced, the bacteria’s cell senescence may be accelerated, and its functionality may be lowered [30]. Ultimately, the quality of natto could be compromised.

Fermentation temperature is one of the important parameters affecting fermentation. The suitable growth temperature for *Bacillus subtilis* and *Bifidobacterium* is around 37 °C. The activity of nattokinase and the sensory score decreased as the fermentation temperature of natto increased (Figure 3(a_1_,b_1_,c_1_)). The increasing temperature, on the one hand, causes the water in the soybeans to evaporate more quickly. The surface layer of the beans becomes dry, affecting mucus formation and, as a result, the taste. On the other hand, we used a 250 mL flask, which may have led to fermentation, as the heat could not be properly dissipated over time, and an accumulation of heat formed in the core. The high temperature impacts strain development and enzyme reactions, causing uneven natto fermentation and hence affecting nattokinase synthesis [31]. pH increased with the increasing temperature (Figure 3(b_1_)). The amine bases produced by the breakdown of soybeans and the organic acids produced by BZ25 are the primary sources of the pH changes. The pH of mixed-bacteria fermented natto was practically the same as that of single-bacteria fermented soybean when the temperature was greater than 37 °C, indicating that the effect of BZ25 is not noticeable after 37 °C. It is possible that the high temperature prevents BZ25 from growing. BZ25’s ability to generate acid may be hampered by the higher temperature. The alterations in free amino nitrogen and nattokinase were varied at different temperatures, as shown in Figure 3(c_1_). The generation of nattokinase was inhibited by high temperatures, while free amino nitrogen levels increased dramatically. The breakdown of soybean protein by excess protease was the predominant source of free amino nitrogen. As a result, high temperatures suppress nattokinase expression while having little to no influence on protease activity.

One of the determining factors of product quality was fermentation time. As the fermentation duration increased (Figure 3(a_2_,b_2_,c_3_)), sensory score decreased, and nattokinase activity and free amino nitrogen increased. It could be because the bacteria in the fermentation reached a steady state of productivity. The catalytic biological reaction did not cease and the fermentation products continued to increase. Nevertheless, longer fermentation times will cause bacteria to enter senescence, resulting in the accumulation of hazardous compounds [32]. As a result, choosing the right fermentation period can help reduce nutrient waste and the formation of toxic metabolites.

After-ripening time can help to minimize natto’s unpleasant smell and make it taste richer and fuller. A longer after ripening time will degrade the product’s quality and freshness, resulting in a lower sensory score (Figure 4). As a result, selecting the most appropriate after-ripening time is critical.

Through the PB test (Appendix A and Table 3) and response surface test (Appendix A), it was determined that fermentation time, fermentation temperature, and inoculation amount were significant factors affecting natto fermentation. PB test and response surface test can effectively be used to estimate the effect of fermentation time, fermentation temperature, and inoculation amount, and their interactions on the natto. According to the findings of the study, the optimal fermentation conditions for achieving the finest natto quality were as follows: soybean 50 g/bottle, NaCl was 0%, sucrose 1%, ratio of GUTU09 to BZ25 1:1, inoculum 7%, fermentation temperature 35.5 °C, and fermentation time 24 h. The nattokinase activity in the above conditions was 144.83 ± 2.66 FU/g, which was higher than the nattokinase activity in the single-bacteria fermentation procedure. The level of free amino nitrogen was 7.02 ± 0.69 mg/Kg. Furthermore, these optimal conditions resulted in an overwhelming sensory score, with the mixed bacteria fermentation natto scoring much higher on appearance, stringiness, flavor, taste, and chewiness than the non-optimized natto.

Under the optimal circumstances outlined above, the thrombolytic action of natto was investigated (Table 4, Figure 7(A_1_,A_2_)). It indicates that as fermentation progressed, strains produced more nattokinase and its thrombolytic activity was improved. The antithrombotic effect includes two aspects: one is the fibrinolysis of the formed thrombus, which is evaluated by the fibrin degradation ability (nattokinase activity); the other is the anticoagulant activity during the formation of fibrin, that is, the inhibition ability of fibrin on the original coagulation [17]. The fibrinolytic activity and anticoagulant activity may reflect the thrombolytic ability to some extent [33].

Single-bacteria fermentation had higher enzyme activity for the first 18 h, but after that, mixed-bacteria fermentation had higher enzyme activity than single-bacteria fermentation (Figure 7B). When both *Bifidobacterium* BZ25 and *Bacillus subtilis* GUTU09 are inoculated at the same time, BZ25 growth is impeded due to the presence of oxygen in the soybean medium at the start of fermentation [34]. According to Hosoi et al. [35], the viability of Lactobacilli in the presence of Bacillus was greatly improved, and they speculated that the production of subtilisin and catalase may play a part in this improvement. Another explanation for the improvement was that the growth of *B. subtilis* GUTU09 in solid-state substrates used dissolved oxygen, allowing *Bifidobacterium* BZ25 to flourish [36]. Simultaneously, the released protease assisted in the hydrolysis of soy protein, providing a nitrogen source for BZ25. Its amylase and other enzymes degraded the polysaccharide in soybeans to give BZ25 a carbon source. These factors promoted the growth of BZ25 [37]. Furthermore, *Bifidobacterium* growth may produce certain compounds that aid Bacillus metabolism. As a result, BZ25 and GUTU09 boosted each other more than single-bacteria fermentation, which had more advantages and was more favorable for bioactive ingredient accumulation. Our experiments revealed that adding a diluent to a natto extract prevented fibrin production, and that the anticoagulant action increased as the enzyme concentration increased (Figure 7C). Thus, mixed-bacteria fermentation methods result in greater nattokinase activity, free amino nitrogen concentration, and sensory assessment. As a result, it is a nutritious food that is readily available to the general public.

## 5. Conclusions

Mixed fermentation increased nattokinase activity, free amino nitrogen and the sensory score of natto. Substantial thrombolytic and anticoagulant effects were also observed. A healthy fermented natto with good sensory characteristics was studied and developed.

## Figures and Tables

**Figure 1 foods-10-02547-f001:**
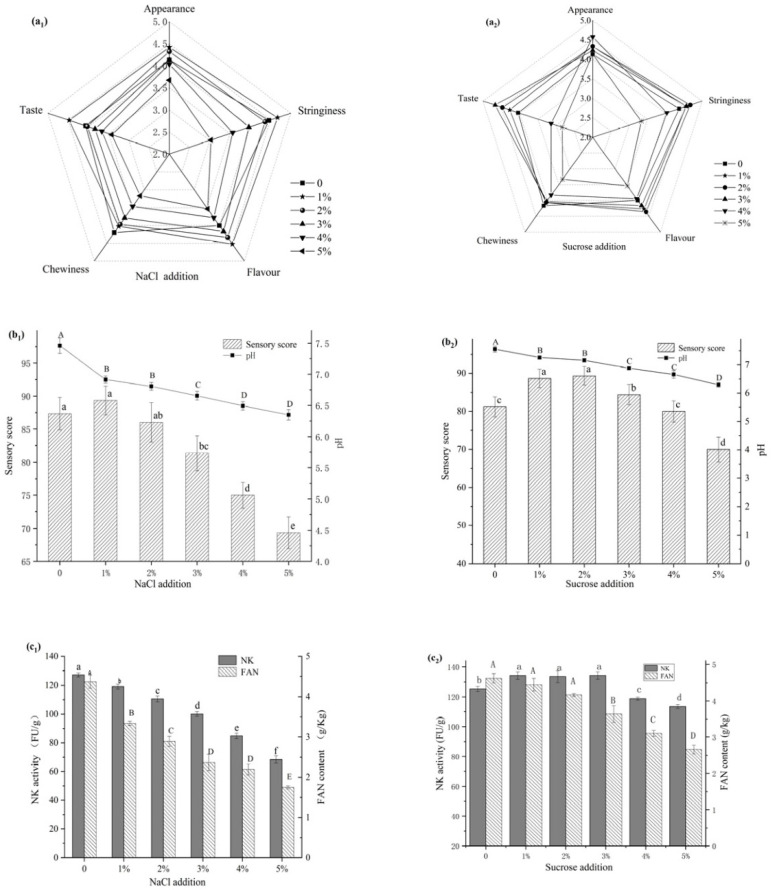
Effects of NaCl addition (**a_1_**,**b_1_**,**c_1_**) and sucrose addition (**a_2_**,**b_2_**,**c_2_**) on natto fermentation. Different letters above each bar means significant differences.

**Figure 2 foods-10-02547-f002:**
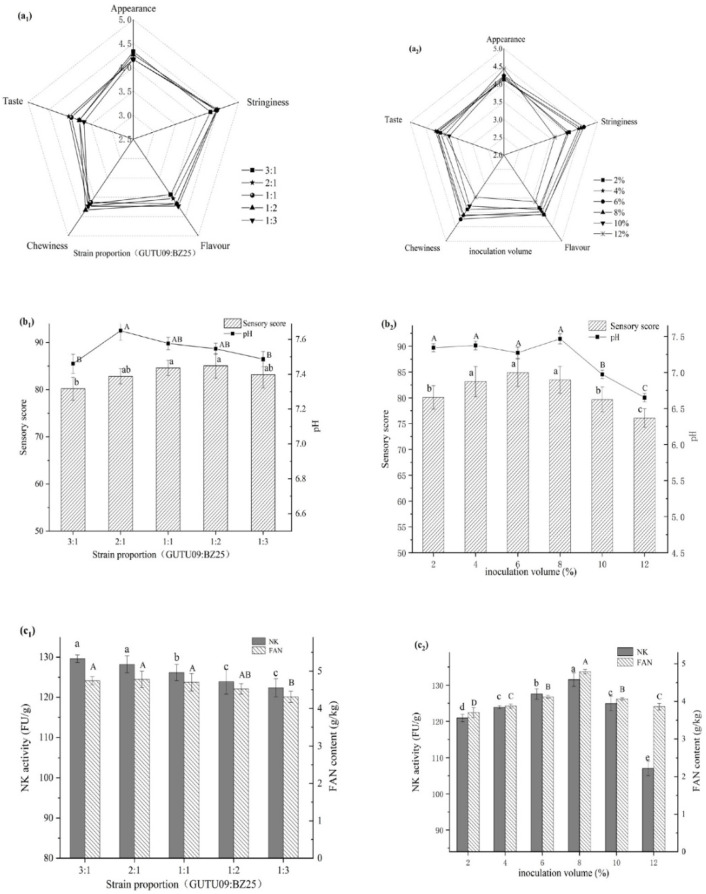
Effects of strain proportion (**a_1_**,**b_1_**,**c_1_**) and inoculation volume (**a_2_**,**b_2_**,**c_2_**) on natto fermentation. Different letters above each bar means significant differences.

**Figure 3 foods-10-02547-f003:**
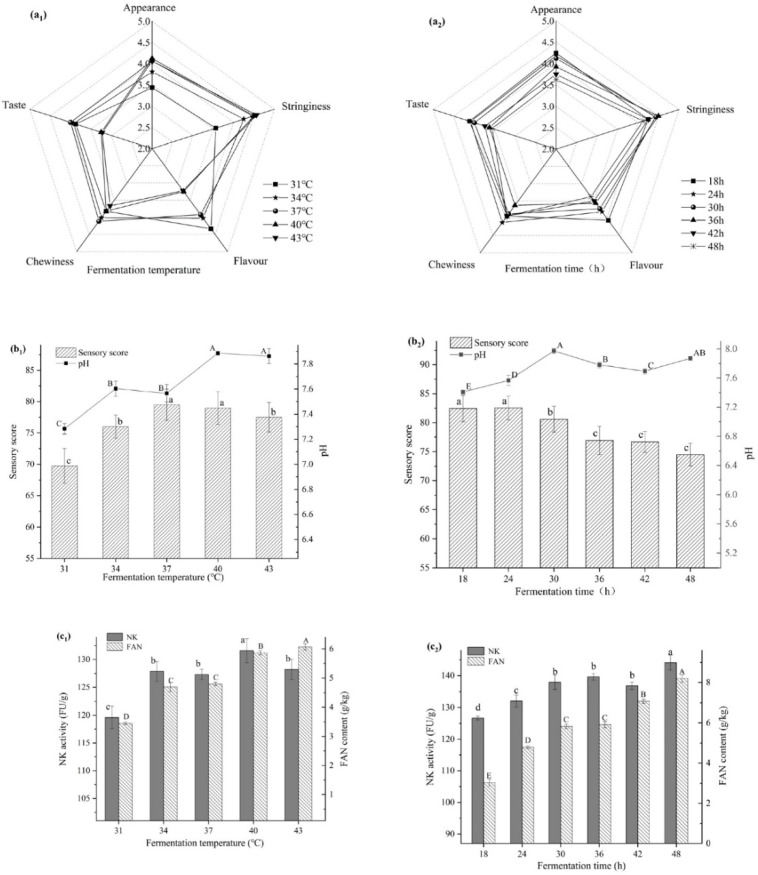
Effects of fermentation temperature (**a_1_**–**c_1_**) and fermentation time (**a_2_**–**c_2_**) on natto fermentation. Different letters above each bar means significant differences.

**Figure 4 foods-10-02547-f004:**
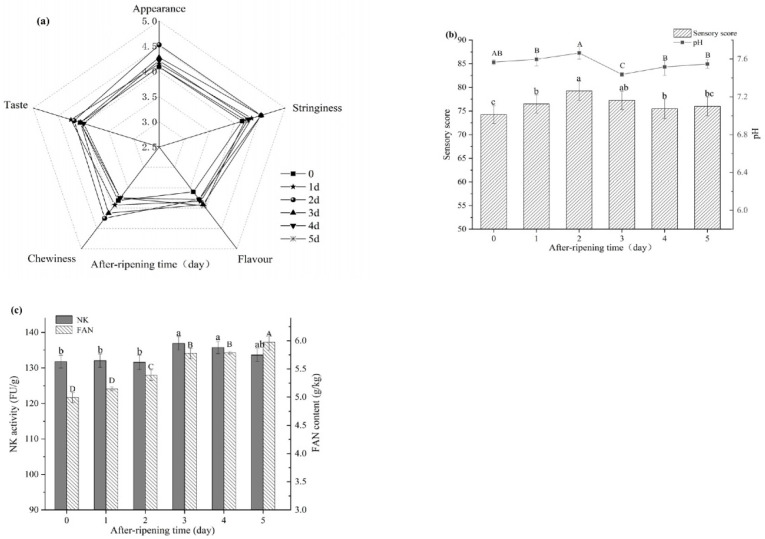
Effects of after-ripening time on natto (**a**–**c**). Different letters above each bar means significant differences.

**Figure 5 foods-10-02547-f005:**
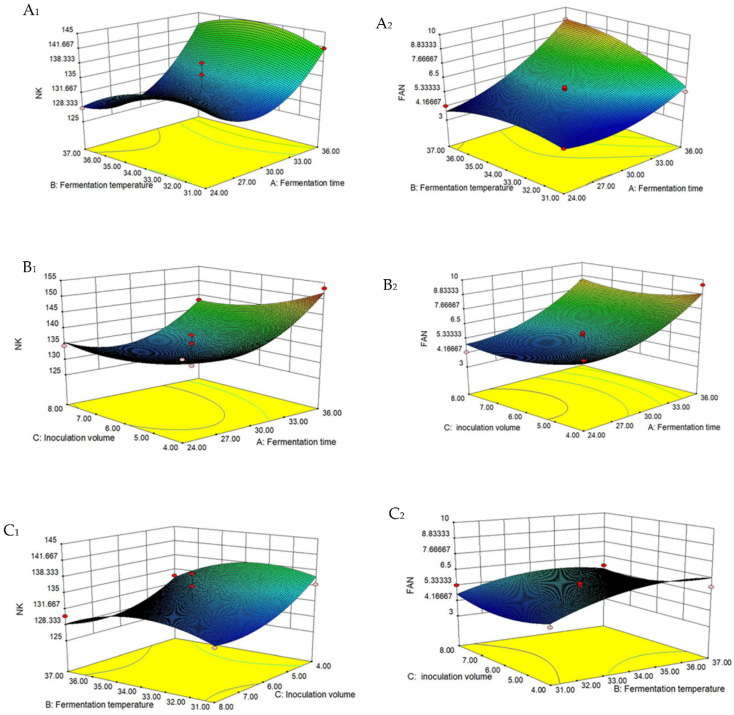
Response surface diagram of NK activity and FAN content as a response to the interaction of: (**A_1_**,**A_2_**) fermentation temperature and fermentation time (inoculation amount = 6%), (**B_1_**,**B_2_**) inoculation amount and fermentation time (temperature = 34 °C), (**C_1_**,**C_2_**) inoculation amount and fermentation temperature (time = 30 h).

**Figure 6 foods-10-02547-f006:**
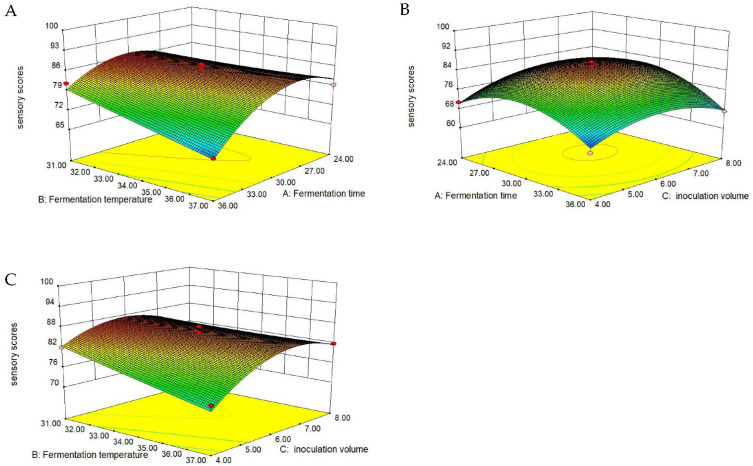
Response surface diagram of sensory score as a response to the interaction of: (**A**) fermentation temperature and fermentation time (inoculation amount = 6%), (**B**) inoculation amount and fermentation time (temperature = 34 °C), (**C**) inoculation amount and fermentation temperature (time = 30 h).

**Figure 7 foods-10-02547-f007:**
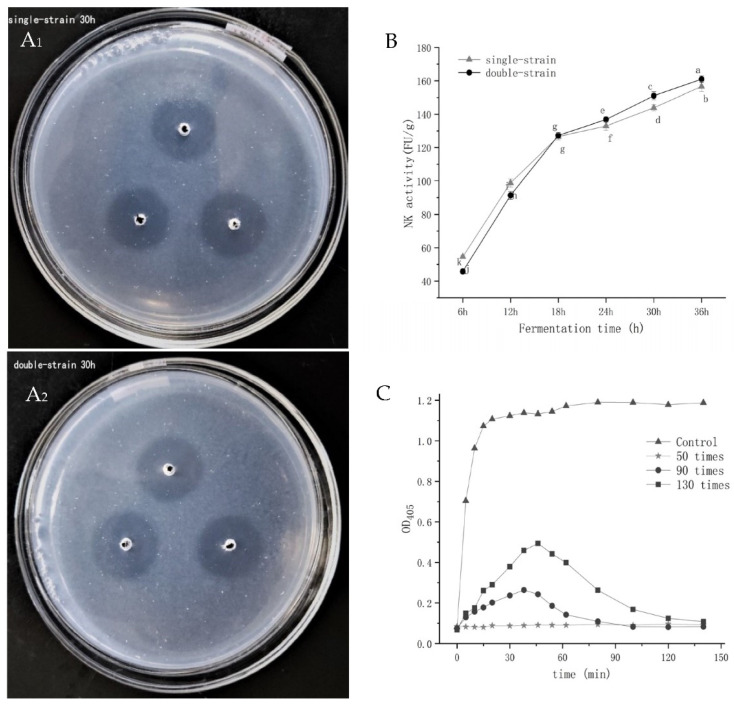
(**A_1_**,**A_2_**_)_ Comparison of the effect of single-bacteria and mixed-bacteria fermented natto on fibrin plate dissolution circle; (**B**) changes in nattokinase activity during fermentation; (**C**) prothrombin condensation inhibition ability.

**Table 1 foods-10-02547-t001:** Minimum and maximum range of the parameters selected in the PB design.

Variables	Units	Levels	
		1	−1
A—NaCl content	%	0	1
B—Sucrose addition	%	1	3
D—Fermentation temperature	°C	37	34
E—Fermentation time	h	30	24
G—strain proportion		2:1	1:1
J—inoculation volume	%	8	6
L—after-ripening time	d	3	1
C, F, H, K		1	−1

**Table 2 foods-10-02547-t002:** Coded and real values of variables in the BBD.

Variables	Units	Levels		
		−1	0	1
D—Fermentation temperature	°C	31	34	37
E—Fermentation time	h	24	30	36
J—Inoculation volume	%	4	6	8

**Table 3 foods-10-02547-t003:** Experimental analysis of variance-response values was based on nattokinase activity, amino acid nitrogen, and sensory score.

Source	Nattokinase Activity						Amino Acid Nitrogen						Sensory Scores					
	Sum of Squares	df	Meansquare	F-Value	*p*	Significance	Sum of Squares	df	Meansquare	F-Value	*p*	Significance	Sum of Squares	df	Meansquare	F-Value	*p*	Significance
Model	492.89	7	70.41	8.92	0.0258	*	29.87	4	7.47	31.9	0.0001	**	280.06	3	93.35	13.10	0.0019	**
A—NaCl content	118.94	1	118.9	15.07	0.0178	*	17.74	1	17.74	75.77	<0.0001	**						
B—Sucrose addition	131.87	1	131.9	16.71	0.015	*	5.45	1	5.45	23.30	0.0019	**						
D—Fermentation temperature	88.35	1	88.35	11.19	0.0287	*	3.19	1	3.19	13.64	0.0077	**	55.04	1	55.04	7.72	0.024	*
E—Fermentation time	62.38	1	62.38	7.9	0.0482	*	3.49	1	3.49	14.9	0.0062	**	101.5	1	101.5	14.24	0.0054	**
G—Strain proportion	28.34	1	28.34	3.59	0.131													
J—Inoculation volume	9.08	1	9.08	1.15	0.3438								123.52	1	123.52	17.33	0.0032	**
L—After-ripening time	53.93	1	53.93	6.83	0.0592													
Residual	31.57	4	7.89	8.92	0.0258		1.64	7	0.23				57.02	8	7.13			
sum	524.46	11		15.07			31.51	11					337.08	11				

Note: ** indicates *p* < 0.01, * indicates *p* < 0.05. Statistical analysis was performed using test design experts.

**Table 4 foods-10-02547-t004:** Changes in fibrinolytic activity during fermentation.

Center Hole Area 1.98 ± 0.39 mm^2^	GUTU09 Single-Strain		Double-Strain Fermentation	
	Dissolving Circle Diameter (mm)	Thrombolytic Area (mm^2^)	Dissolving Circle Diameter (mm)	Thrombolytic Area (mm^2^)
6 h	5.46 ± 0.45 ^g^	21.53 ± 3.4 ^G^	4.77 ± 0.69 ^g^	16.16 ± 4.75 ^G^
12 h	14.17 ± 0.31 ^f^	155.76 ± 6.55 ^F^	13.75 ± 0.51 ^f^	146.64 ± 10.66 ^F^
18 h	16.13 ± 0.48 ^e^	202.46 ± 11.86 ^E^	16.12 ± 0.45 ^e^	202.11 ± 11.02 ^E^
24 h	17.1 ± 0.20 ^d^	232.64 ± 5.08 ^D^	17.77 ± 0.2 ^c^	245.83 ± 5.18 ^C^
30 h	18.28 ± 0.38 ^b^	272.8 ± 10.66 ^B^	19.70 ± 0.35 ^a^	302.73 ± 10.37 ^A^

Note: Data are mean values of three independent experiments ± standard deviation. Mean values displaying different letters within each row are significantly different according to the Duncan test at 95% confidence level.

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
