# Peer review of "Effect of Fermentation Parameters on Natto and Its Thrombolytic Property"

_foods, 2021, doi:10.3390/foods10112547_

Round 1

Reviewer 1 Report

The authors studied the effect of mixed bacteria fermentation parameters on natto and its thrombolytic property. The study is well designed and only minor revisions are suggested.

Minor revisions 

-The reference of the previous study on which the authors base their hypothesis (lines 12-13) should be included in Introduction section. 

  • Please revise in Italics format the scientific names of bacteria in the text.
  • Section Introduction, line 43, "the nutrition of soybeans". Please revise. An alternative could be: "the nutritional value of soybeans".
  • Section Introduction, line 60, "may used". Please revise to "may be used".
  • Section Introduction, line 66, "will be". Please revise to "was".
  • Section Conclusions, line 483, "effectively use". Please revise to "effectively be used".
  • Section Conclusions, lines 492-493, "As well as substantial thrombolytic and anticoagulant effects". Please revise. An alternative could be "Substantial thrombolytic and anticoagulant effects were also observed".
  • Section Conclusions, lines 493-494-495. Please revise the sentence "In conclusion, ... sensory evaluation". The verb is missing.

Reviewer 2 Report

This paper describes the effect of fermentation parameters on the thrombolytic activity (due to nattokwinase) of natto. The approach is rather descriptive in nature, with very little attempts to formulate and test scientific hypotheses. I suggest separating the results and discussion section. This would avoid some repetition of content.

Some sections do not have any discussion only a presentation of the results.

Minor

Abstract

l11 I doubt it is popular because it contains bioactive compounds; do you have evidence for that?

l12 mixed with? with Bifidobacterium added? Which species?

l13 sensory?

l16-17 quality measured how? sensory? functional?

l17 rather than suitable, I would write optimal; I am stopping language revision here; there are too many problems with English usage; please have the manuscript read by a native English speaker or by a proof-reading service

l21 increased compared to what

Introduction

l39 here and elsewhere, species and genera in italics

l41 Natto produces? Please rephrase

l50 ammonia

l53 very unclear? does nattokwinase activity appear on the label?

Materials and methods

l69 prepared how, please describe

l75 cultivated how? Aerobically, anaerobically? With pH control?

l79 adjusted how

l83 a certain amount?

l88 initial pH?

l91 what do you mean by single factor experiments?

l131 frowned upon?

l134-135 why?

Results

figure 1 define NK and FAN in the legend of figure 1; are these the single factor experiments you mentioned in methods?

l175-176, technically this may be revealed by a statistical analysis; ANOVA? one-way?

l180 increased by how much?

l189-190 this is frankly surprising. Did you measure dissolved oxygen concentration? It might well be that O2 got depleted and this favoured the growth of the Bifidobacterium (which should be oxygen sensitive) and inhibited the strictly aerobic Bacillus strain

l200-201 I am not sure this is the reason, see above

l216-217 the beans have been treated at 121°C, they cannot have any metabolism by themselves

l219-220: well, so inoculum is confounded with water content. You should amend any discussion accordingly; a better strategy would have been to use different inocula based on concentrated cell biomass delivered in the same amount of water...

l236-239 this depends on several other factors (size and shape of the container, temperature control...)

l261 age?

l285 what do you mean by "filtered out"?

l288 which 3 models? Do you mean factors?

Table 3 can go in supplementary material; PB designs are orthogonal designs; you can safely remove effects with a p value for the t test >0.05; I assume you corrected for multiple testing. How? Bonferroni? Holm-Bonferroni?

Table 5 and 6 can go in supplementary (the figure is enough).

Reviewer 3 Report

In this paper, Yang et al.  investigated the effects of different fermentation parameters on the quality of natto by the Bacillus subtilis GUTU09 and Bifidobacterium BZ25 to determine the fermentation parameter using Plackett-Burman Design in conjunction with Response Surface Methodology.

Scientific level of the paper is good and statistical approach is fine. However, the article could be improved by simplifying the presentation and carrying out complete editing of the article by  a native speaker of English. In addition, i suggest to rewrite the very long sentences throughout the entire manuscript.

Round 2

Reviewer 3 Report

In this revised manuscript, the authors have addressed all the comments of the reviewer and revised the paper accordingly. Therefore, the article is suitable for publication in Foods.